# *APOE* Polymorphism and Endocrine Functions in Subjects with Morbid Obesity Undergoing Bariatric Surgery

**DOI:** 10.3390/genes13020222

**Published:** 2022-01-25

**Authors:** Per G. Farup, Aina Jansen, Knut Hestad, Jan O. Aaseth, Helge Rootwelt

**Affiliations:** 1Department of Research, Innlandet Hospital Trust, PB 104, N-2381 Brumunddal, Norway; knut.hestad@inn.no (K.H.); Jan.Aaseth@inn.no (J.O.A.); 2Department of Clinical and Molecular Medicine, Faculty of Medicine and Health Sciences, Norwegian University of Science and Technology, N-7491 Trondheim, Norway; 3Department of Surgery, Innlandet Hospital Trust, N-2819 Gjøvik, Norway; Aina.Jansen@sykehuset-innlandet.no; 4Department of Health and Nursing Science, Faculty of Health and Social Sciences, Inland Norway University of Applied Sciences, N-2418 Elverum, Norway; 5Department of Medical Biochemistry, Oslo University Hospital, N-0424 Oslo, Norway; hrootwel@ous-hf.no

**Keywords:** *APOE* polymorphism, endocrine function, obesity, bariatric surgery

## Abstract

Background: Obesity is an interplay between genes and the environment, including lifestyle. The genetics of obesity is insufficiently understood. Apolipoprotein E (*APOE*) genetic polymorphism has been associated with a wide range of disorders. Knowing that some *APOE* alleles are associated with obesity and endocrine disorders that are common in obesity, the present study aimed at exploring associations between *APOE* polymorphisms and endocrine functions in subjects with obesity undergoing bariatric surgery. Methods: Analyses of hormones in blood collected before and one year after bariatric surgery were examined. The *APOE* alleles were grouped as follows: E2 = ε2ε2 + ε2ε3; E3 = ε3ε3 + ε2ε4; E4 = ε3ε4 + ε4ε4. The *APOE* groups were analysed as nominal and ordered groups (E2-E3-E4) with a linear mixed model to predict the hormonal effects of the groups. Results: Forty-nine women (79%) and thirteen (21%) men with a mean age of 47.7 (SD 8.5) years were included in the study. The adiponectin level was significantly lower (*p* < 0.05) in the E2 group compared with the E4 group. Adiponectin and cortisol were positively and negatively associated, respectively, with the ordered *APOE* groups. Conclusions: The ordered *APOE* groups E2-E3-E4 were significantly associated with high and low levels of adiponectin and cortisol, respectively. The findings indicate *APOE*-mediated effects on body weight and metabolic functions in subjects with morbid obesity.

## 1. Introduction

The cause of obesity and overweight is an energy imbalance between calories consumed and expended [1]. Lifestyle factors such as unfavourable dietary habits, a sedentary lifestyle and psycho-social factors are most important, but endocrine disorders and genetic predisposition also have an identifiable contribution to obesity [2].

Ghrelin and the adipokines leptin and adiponectin have appetite-regulating effects [3]. Adiponectin is a metabolic health product of the adipose tissue and exerts appetite-regulating effects [4]. Leptin is released from adipose tissue and regulates appetite and energy homeostasis [5]. Leptin deficiency is a primary cause of obesity [2,5]. Leptin and ghrelin, together with other hormones, regulate hunger and satiety [3,6].

Endocrine disorders are common in subjects with obesity. Cushing’s syndrome, growth hormone deficiency, hypothyroidism and pseudohypoparathyroidism are associated with obesity and are secondary causes of obesity [2]. Sex hormones are influenced by obesity, both in men and women, and tend to normalise after bariatric surgery [7,8].

Obesity is an interplay between genes, epigenetics and the environment, including lifestyle [9]. Genes’ regulation of obesity is either monogenic or polygenic and is often poorly understood. Apolipoprotein E (*APOE*) genetic polymorphism has been associated with a wide range of disorders. A recently published study reported associations with 37 outcomes representing 18 distinct disorders [10]. *APOE* ε4ε4 was associated with a reduced prevalence of obesity (OR = 0.78), and ε2ε2 with an increased prevalence of obesity (OR = 1.22) compared with ε3ε3. Apolipoprotein E could be the link between *APOE* genetic polymorphism and obesity. The conventional conception has been that apolipoprotein E promotes obesity by the increase in lipid to white adipose tissue. More recently, apolipoprotein E3 in the brain has been associated with fat accumulation and obesity, and hepatically expressed apolipoprotein E3 has the opposite effect [11]. In a previous study, we demonstrated an association between the presence of the allele ε2 and improvement in all psychosomatic disorders, and positive and negative associations between c-reactive protein (CRP) and the alleles ε2 and ε4, respectively [12].

Knowing that some *APOE* alleles are associated with obesity and that endocrine disorders are common in subjects with obesity, the present study aimed to explore associations between *APOE* polymorphism and endocrine functions in subjects with obesity undergoing bariatric surgery.

## 2. Materials and Methods

### 2.1. Study Design and Participants

This retrospective cohort study used data from the prospective cohort study MO-BiPS (Morbid Obesity—BioPsychoSocial impacts). The data are subsets of data used for other purposes in previously published papers [12,13].

Subjects with morbid obesity, defined as BMI > 40 kg/m^2^ or BMI > 35 kg/m^2^ with obesity-related complications and referred to the obesity unit at Innlandet Hospital Trust, Gjøvik, Norway for evaluation of bariatric surgery, were available for inclusion. Subjects with present or previous somatic or psychiatric disorders or drug or alcohol abuse were excluded at the doctors’ discretion.

In the first six months after inclusion, the participants went through a behavioural weight loss intervention with three separate one-hour-long individual consultations and seven weekly group meetings with information about the operation, dietary recommendations and physical activity advice. During the last three weeks prior to bariatric surgery, they followed a strict “crispbread diet” or an alternative powder diet, both with a maximum of 4200 kJ [14]. Bariatric surgery was performed as a standard Roux-en-Y gastric bypass or gastric sleeve procedure at the surgeons’ discretion [15,16]. There were regular follow-up visits after surgery.

Data from the visits immediately before surgery and the follow-up visit one year later were used in this study. Participants with complete datasets from these visits were included in the analyses.

### 2.2. Variables

This study used the following data from the prospective MO-BiPS cohort study:Demographic and anthropometric data: age (years), gender (male/female), height (m), body weight (kg) and body mass index (BMI; kg/m^2^);Blood tests (reference values are given for the age and BMI of the study population);Thyroid Stimulating Hormone (TSH); reference values women/men: 0.5–3.6/0.27–4.2 mIE/L;Free Thyroxin (fT4); reference values women/men: 8.0–21/12–22 pmol/L;Free Triiodothyronine (fT3); reference values women/men: 2.8–7.0/2.7–6.3 pmol/L;Reverse Triiodothyronine (rT3); reference values women/men: 0.23–0.54/0.14–0.54 pmol/L;Parathyroid hormone (PTH); reference values 1.5–7.0 pmol/L;Cortisol; reference morning values: 138–690 nmol/L;Sex Hormone Binding Globulin (SHBG); reference values women/men: 23–100/8–60 nmol/L;Testosterone; reference values: women < 50 years of age: ≤1.9; ≥50 years of age: ≤1.1 nmol/L, men 20–40 years: 7.2–24; ≥40 years: 4.6–24 nmol/L;Free Testosterone Index (FTI) (testosterone (nmol/L) × 10/SHBG (nmol/L)): women 0.1–0.3; men age 20–39: 2.4–11.8; age 40–64: 1.5–7.3;HbA1c; reference values 4.7–6.0%.

Jansen et al. described the hormone analyses in detail [12].

Leptin; reference values for BMI < 25: women/men: 80–2500/< 950 pmol/L;Adiponectin; reference values women/men 4–22/2–20 mg/L;Ghrelin (acylated and non-acylated); reference values: 686–1731 pg/mL.

The adipokines were analysed at the Hormone Laboratory, Oslo University Hospital, Oslo, Norway (accredited by the Norwegian Accreditation according to the requirements of the NS-EN ISO/IEC 17,025 TEST 099) with competitive radioimmunoassay (RIA). Ghrelin, leptin and adiponectin were not included in the accreditation. The results were used in another context in a recently submitted paper by Jansen A. et. al.

Genotyping. The *APOE* alleles were determined. There is no consensus regarding the grouping and order of the *APOE* variants. In particular, grouping of the ε2ε4 is difficult. The following order of the *APOE* variants was chosen: ε2ε2, ε2ε3, ε3ε3, ε2ε4, ε3ε4, e4ε4, which is in accordance with other papers, e.g., the comprehensive review by Lumsden et al. [10]. Based on the order of the *APOE* variants and because some groups were small (the smallest one was ε2ε4 with one participant), we combined the genetic variants into three groups: E2 (ε2ε2 + ε2ε3), E3 (ε3ε3 + ε2ε4) and E4 (ε3ε4 + ε4ε4) according to Han et al. [17]. This is in accordance with the observation that for many clinical or biochemical properties investigated and published [10,12,17], the ε2 and ε4 alleles have opposite effects relative to the ε3 allele. Therefore, it makes sense in a clinical setting to group ε2ε4 alleles phenotypically as E3 although the ε2ε4 genotype is not genetically ε3.

Genotyping was performed at the Department of Medical Biochemistry, Oslo University Hospital—Rikshospitalet, Oslo, Norway using real-time polymerase chain reactions, described by Hestad et al. [18].

### 2.3. Statistical Analyses

Descriptive data are reported as mean (SD) and number with proportion (%). Values before and after surgery were compared using a paired *t*-test. A linear mixed regression model for repeated measurements was used to analyse predictors of the hormones, and the results were reported as the estimate (B-value) with 95% confidence interval and *p*-values. The *APOE* groups E2, E3 and E4 were analysed as nominal groups with comparisons between the groups one-to-one and as an ordered variable with rank order: E2-E3-E4. *p*-values < 0.05 were judged as statistically significant. The normality of the residuals of the mixed model linear regression was confirmed by the visual inspection of QQ-plots. The analyses were performed with IBM SPSS Statistics for Windows, version 27.0 (IBM Corp., Armonk, NY, USA). The *p*-values were adjusted for multiple testing ad modum Benjamini–Hochberg with R-Studio version 1.4.1106 and reported as q-values. Because the results showed associations between adiponectin and the *APOE* genetic variants, HbA1c was added post hoc as a dependent variable.

## 3. Results

One hundred and forty-two subjects were included in the main study, twenty-one withdrew during the behavioural intervention and one hundred and twenty-one underwent bariatric surgery. Sixty-two participants who had complete data before and 12 months after surgery were included in this study. Table 1 gives the patient characteristics.

There was a significant difference in adiponectin between the groups E2 and E4, differences in HbA1c (post hoc analyses) between E2 and E3 and differences between E3 and E4 adjusted for age, gender, BMI and point of time. No significant associations were detected after adjusting for multiple testing. Table 2 gives all associations between the hormones and the *APOE* groups.

Table 3 gives the associations between the ordered E groups (order E2-E3-E4) and the hormones. Adiponectin and cortisol were positively and negatively associated with the ordered E groups. No significant associations were detected after adjusting for multiple testing.

Figure 1 shows the two significant associations in Table 3 between the *APOE* groups and the hormones presented as the regression lines adjusted for the mean of the other variables.

## 4. Discussion

In our current study, the main findings include the positive and negative associations between the ordered *APOE* (E2-E3-E4) and adiponectin and cortisol, respectively. No significant associations were observed between the *APOE* alleles and the other hormones.

Adiponectin is a hormone that passes into the brain and has an appetite-regulating effect, increases energy expenditure and thermogenesis, potentiates the effect of leptin and reduces body weight [4,19]. Adiponectin values are low in subjects with obesity [4]. In healthy humans, adiponectin increases by overnutrition and weight gain [20]. Because adiponectin resistance rapidly occurs after fatty acid intake, the rise might be a compensatory response [21]. In subjects with obesity and insulin resistance, adiponectin does not respond to overfeeding, indicating adiponectin resistance [4,21]. The positive association between E4 and adiponectin in this study is in accordance with the protective effect of ε4 against obesity [10].

In addition to the appetite-regulating effects, adiponectin is an indicator of insulin sensitivity. The positive association between E4 and adiponectin indicates a protective effect of ε4 on diabetes, as reported in the meta-analysis by Lumsden et al. [10]. Our findings on adiponectin led to post hoc analyses of HbA1c. The significantly higher HbA1c values in the E4 group compared with the E3 group is not as expected if ε4 protects against diabetes, and is not as reported by Lumsden et al. [10]. HbA1c is a commonly used marker for the diagnosis and monitoring of treatment in diabetes. Without detailed knowledge of the treatment given to subjects with diabetes and the blood glucose levels of the subjects in the different groups, the HbA1c results are unreliable as a marker of insulin sensitivity.

Adiponectin has also been associated with unfavourable effects, such as chronic heart failure, chronic kidney disease, cognitive impairment and Alzheimer’s disease [22]. The discrepancy between the beneficial effects on glucose metabolism, inflammation and atherosclerosis, and no direct effect on the risk of type II diabetes and cardiovascular disease, and increased mortality across several diseases, has been mentioned as the adiponectin paradox [23].

Screening for endocrine disorders such as hypogonadism, polycystic ovary syndrome, Cushing’s syndrome, thyroid dysfunction, diabetes, growth hormone deficiency and pseudohypoparathyroidism is mandatory in subjects with severe obesity [2,24]. In this study, cortisol was the only endocrine function associated with *APOE* polymorphism. High cortisol values are associated with overweight [25]. The negative association between the E4 group and cortisol seen in this study indicates that the protective effect of ε4 against obesity could in part be related to corticosteroid metabolism. The meta-analysis by Lumsden et al. found no association between *APOE* polymorphism and disorders of the adrenal glands [10].

Glucocorticosteroids inhibit adiponectin secretion in animals, healthy persons and non-obese subjects with Cushing’s syndrome [26]. The inhibitory effect could explain the negative association between adiponectin and cortisol in this study. Obesity has been reported to mask this effect, which seemed not to be the case in this study [26].

In accordance with other studies, no statistically significant associations of *APOE* polymorphism with thyroid and parathyroid disorders, sex hormones, and the adipokines ghrelin and leptin were observed [10]. In a previous study, we did not see any association between *APOE* polymorphism and body weight and weight loss after surgery [12]. In the same study, there was a fall in CRP after treatment; ε2 and ε4 were significantly associated with high and low CRP, respectively. The associations between *APOE* genotypes and endocrine functions could indicate that change in weight is a metabolic effect and not a direct *APOE* genotype effect.

### Strengths and Limitations

The study population was an unselected and representative group of subjects referred for evaluation of bariatric surgery. The treatment followed national and international guidelines with a lifestyle intervention before surgery, which was performed with standard techniques, and regular follow-ups after surgery. A broad spectrum of hormones known to be related to obesity, or possibly related to obesity, was analysed. Information about the lipid profiles would have been desirable.

The unbalanced gender distribution in favour of females limits the external validity. All the analyses were adjusted for gender, and the limited number of participants made gender-specific analyses inappropriate. The results might represent female-specific effects.

There is no consensus about the grouping of the *APOE* variants, and this depends on the study population. Instead of having each individual genotype ε2ε2, ε2ε3, ε3ε3, ε2ε4, ε3ε4 and e4ε4 as separate groups, we decided to combine genotypes pairwise into three groups: E2 (ε2ε2 + ε2ε3), E3 (ε3ε3 + ε2ε4) and E4 (ε3ε4 + ε4ε4) to obtain larger groups whilst ensuring clear E2 and E4 profiles. A different grouping or leaving out the participant with ε2ε4 could have given other results but has not been performed.

The explorative design with multiple testing increases the possibilities of type I errors. To reduce type I errors, *p*-values were adjusted for multiple testing. This strengthening of the methods, which removed all significant associations, demonstrates the uncertainty of the results. Similarly, because of the relatively small sample size, particularly in the *APOE* subgroups, negative results do not exclude an effect (type II error). Discrepancies between this and other studies could also be due to the study population (i.e., subjects with morbid obesity), different ethnic groups and the study design. The results could be specific for subjects with obesity. The lack of a reference group with healthy subjects is a limitation. Nevertheless, the results indicate endocrine effects of clinical importance of *APOE* polymorphism in subjects with morbid obesity. Confirmations from larger studies are desirable.

## 5. Conclusions

*APOE* polymorphism seems to have endocrine effects of clinical significance in subjects with morbid obesity. The ε4 allele was positively and negatively associated with adiponectin and cortisol, respectively, indicating effects on metabolic functions and thus body weight.

## Figures and Tables

**Figure 1 genes-13-00222-f001:**
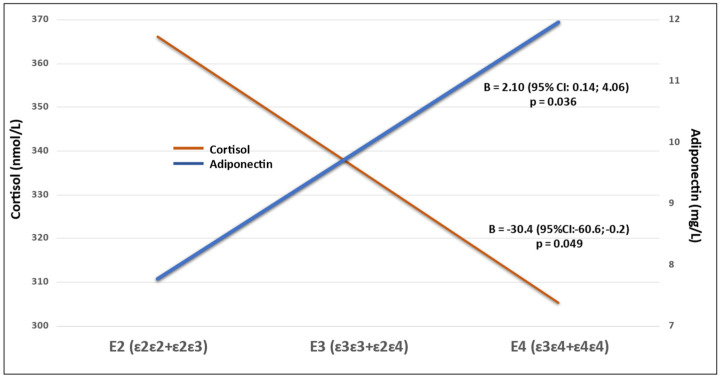
The associations between the ordered *APOE* groups and cortisol and adiponectin presented as the regression lines adjusted for the mean of age, gender, BMI and point of time.

**Table 1 genes-13-00222-t001:** Patient characteristics. For gender-dependent variables, values are given for men and women.

	Before Surgery	After Surgery	Statistics *p*-Values
Male/female	13 (21%)/49 (79%)	---	---
Age (years)	44.7 (8.5)	---	---
Type of surgery R-Y Gastric bypass/Sleeve	52 (84%)/10 (16%)	---	---
*APOE* groups ε2ε3/ε3ε3/ε2ε4/ ε3ε4/ε4ε4	7 (11%)/31 (50%)/1 (2%)/ 21 (34%)/2 (3%)		
BMI (kg/m^2^)	39.0 (3.2)	28.2 (3.6)	<0.001
TSH (mIE/L)	1.71 (1.39)	1.66 (0.98)	0.764
Thyroxin (T4) (pmol/L)	16.8 (3.5)	15.9 (3.9)	0.089
Free T3 (pmol/L)	4.58 (0.76)	4.59 (1.12)	0.959
Reverse T3 (pmol/L)	0.51 (0.20)	0.41 (0.17)	<0.001
Parathyroid hormon (pmol/L)	5.51 (2.19)	5.61 (1.77)	0.721
Cortisol (nmol/L)	296 (86)	358 (105)	<0.001
SHBG (nmol/L) Men Women	49.3 (29.8) 31.5 (12.4) 54.1 (31.3)	82.4 (42.2) 56.15 (28.4) 89.4 (42.8)	<0.001 0.002 <0.001
Testosterone (nmol/L) Men Women	3.2 (5.5) 12.6 (5.7) 0.7 (0.3)	4.9 (8.9) 20.4 (8.5) 0.8 (0.4)	0.001 <0.001 0.294
Free testosterone index Men Women	1.03 (1.79) 4.24 (1.43) 0.17 (0.11)	0.96 (1.88) 4.21 (1.86) 0.10 (0.06)	0.570 0.968 <0.001
Leptin (pmol/L) Men Women	2212 (837) 1370 (459) 2436 (771)	1107 (674) 529 (433) 1260 (644)	<0.001 <0.001 <0.001
Adiponectin (mg/L) Men Women	7.7 (4.2) 4.4 (1.8) 8.6 (4.2)	13.1 (7.0) 10.8 (5.2) 13.7 (7.4)	<0.001 <0.001 <0.001
Ghrelin (pg/mL) R-Y Gastric Bypass Gastric sleeve	1108 (369) 1091 (367) 1201 (379)	1247 (615) 1375 (590) 582 (73)	0.025 <0.001 <0.001
HbA1c (%)	5.50 (1.00)	5.17 (0.70)	<0.001

**Table 2 genes-13-00222-t002:** The *APOE* groups (one-by-one) as predictors of the hormones adjusted for age, gender, BMI and point of time (before/after surgery) were analysed with linear mixed model. The q-values are Benjamini–Hochberg-adjusted *p*-values for 13 analyses between each *APOE* group and the hormones.

Dependent Variable	*APOE* E3 Compared with E2 B (95%CI) *p*-Value/q-Value	*APOE* E4 Compared with E2 B (95%CI) *p*-Value/q-Value	*APOE* E4 Compared with E3 B (95%CI) *p*-Value/q-Value
TSH (mIE/L)	0.42 (−0.47; 1.31) *p* = 0.350/q = 0.766	0.48 (−0.43; 1.40) *p* = 0.294/q = 0.713	0.06 (−0.51; 0.64) *p* = 0.825/q = 0.894
Thyroxin (T4) (pmol/L)	−0.88 (−3.56; 1.79) *p* = 0.511/q = 0.766	−0.68 (−3.42; 2.07) *p* = 0.623/q = 896	0.21 (−1.52; 1.93) *p* = 0.812/q = 0.894
Free T3 (pmol/L)	0.21 (−0.44; 0.85) *p* = 0.522/q = 0.766	−0.13 (−0.79; 0.53) *p* = 0.688/q = 0.896	−0.34 (−0.75; 0.07) *p* = 0.104/q = 0.451
Reverse T3 (pmol/L)	−0.02 (−0.17; 0.13) *p* = 0.789/q = 0.855	0.01 (−0.14; 0.17) *p* = 0.869/q = 0.938	(−0.06; 0.13) *p* = 0.498/q = 0.809
Parathyroid hormone (pmol/L)	0.49 (−1.02; 2.01) *p* = 0.519/q = 0.766	0.45 (−1.10; 2.01) *p* = 0.563/q = 0.896	−0.04 (−1.02; 0.94) *p* = 0.936/q = 0.936
Cortisol (nmol/L)	−20.7 (−86.5; 45.0) *p* = 0.530/q = 0.766	−56.0 (−123.6; 11.5) *p* = 0.102/q = 0.459	−35.3 (−77.6; 6.9) *p* = 0.100/q = 0.451
SHBG (nmol/L)	19.94 (−6.06; 45.94) *p* = 0.130/q = 0.689	21.94 (−4.77; 48.64) *p* = 0.106/q = 0.459	2.00 (−14.75; 18.76) *p* = 0.812/q = 0.894
Testosterone (nmol/L)	0.48 (−2.01; 2.97) *p* = 0.699/q = 0.826	−0.10 (−2.66; 2.46) *p* = 0.938/q = 0.938	−0.58 (−2.19; 1.02) *p* = 0.469/q = 0.809
Free testosterone index	0.01(−0.53; 0.54) *p* = 0.980/q = 0.980	−0.09 (−0.64; 0.46) *p* = 0.758/q = 0.896	−0.09 (−0.44; 0.25) *p* = 0.596/q = 0.861
Leptin (pmol/L)	−98 (−483; 286) *p* = 0.610/q = 0.793	−248 (−644; 147) *p* = 0.212/q = 0.689	−150 (−398; 97) *p* = 0.227/q = 0.619
Adiponectin (mg/L)	3.02 (−1.22; 7.27) *p* = 0.159/q = 0.689	**4.66 (0.29; 9.02)** * **p** * **= 0.037/q = 0.459**	1.63 (−1.11; 4.37) *p* = 0.238/q = 0.619
Ghrelin (pg/mL)	176 (−209; 562) *p* = 0.363/q = 0.766	73 (−322; 469) *p* = 0.712/q = 0.896	−103 (−351; 145) *p* = 0.410/q = 0.809
HbA1c	**−0.87 (−1.59; −0.15)** * **p** * **= 0.019/q = 0.247**	−0.36 (−1.10; 0.38) *p* = 0.329/q = 0.713	**0.51 (0.04; 0.97)** * **p** * **= 0.034/q = 0.442**

**Table 3 genes-13-00222-t003:** Ordered *APOE* groups (E2-E3-E4) as a predictor of the hormones adjusted for age, gender, BMI and point of time (before/after surgery), analysed with linear mixed model. The q-values are Benjamini–Hochberg-adjusted *p*-values for 13 analyses between the *APOE* order and the hormones.

Dependent Variable	*APOE* Order B (95%CI) *p*-Value/q-Value	Dependent Variable	*APOE* Order B (95%CI) *p*-Value/q-Value
TSH (mIE/L)	0.18 (−0.23; 0.59) *p* = 0.375/q = 0.813	Testosterone (nmol/L)	−0.23 (−1.38; 0.92) *p* = 0.695/q = 0.883
Thyroxin (T4) (pmol/L)	−0.16 (−1.39; 1.07) *p* = 0.797/q = 0.883	Free testosterone index	−0.06 (−0.31; 0.19) *p* = 0.634/q = 0.883
Free T3 (pmol/L)	−0.16 (−0.46; 0.14) *p* = 0.297/q = 0.772	Leptin (pmol/L)	−133 (−310; 43) *p* = 0.136/q = 0.589
Reverse T3 (pmol/L)	0.01(−0.05; 0.08) *p* = 0.664/q = 0.883	Adiponectin (mg/L)	**2.10 (0.14; 4.06)** * **p** * **= 0.036/q = 0.319**
Parathyroid hormone (pmol/L)	0.14 (−0.56; 0.84) *p* = 0.694/q = 0.883	Ghrelin (pg/mL)	−9 (−188; 170) *p* = 0.918/q = 0.918
Cortisol (nmol/L)	**−30.4 (−60.6; −0.2)** * **p** * **= 0.049/q = 0.319**	HbA1c (%)	0.02 (−0.31; 0.40) *p* = 0.815/q = 0.883
SHBG (nmol/L)	8.01 (−4.06; 20.1) *p* = 0.189/q = 0.614		

## Data Availability

The raw datasets generated and analysed during the current study are not publicly available in order to protect participant confidentiality. Case report forms (CRFs) on paper are safely stored. The data were transferred deidentified to SPSS and R-studio for statistical analyses. The data files are stored by Innlandet Hospital Trust, Brumunddal, Norway, on a server dedicated to research. The security follows the rules given by the Norwegian Data Protection Authority, P.O. Box 8177 Dep. NO-0034 Oslo, Norway. The data are available on request to the authors.

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
