# Peer review of "APOE Polymorphism and Endocrine Functions in Subjects with Morbid Obesity Undergoing Bariatric Surgery"

_genes, 2022, doi:10.3390/genes13020222_

Round 1
Reviewer 1 Report
It is a very interesting article in which the authors have examined the associations between APOE allele and endocrine functions in the subjects with obesity undergoing bariatric surgery. The authors reported that significant associations of APOE allele and genotypes with the levels of adiponectin and cortisol, respectively. I think the manuscript would gain interest for the readerships of this journal, Gene if the authors were more precise in the description of the results and discussion. Major revision of the study will be needed to achieve excellence
- Abstract:
Grammar issues
- Line 15, you may need to change to “Knowing that some APOE alleles are associated with obesity and that endocrine disorders are common in obesity” to “Knowing that some APOE alleles are associated with obesity and endocrine disorders that are common in obesity
- Line 17 it’s better to state “APOE gene or polymorphisms” instead of “polymorphism” since there are many polymorphisms in this gene
- Line 20.E3 = É›3É›3+É›2É›4 may be typo since it should be “E3 = É›3É›3+É›3É›4”
- Line 26. It might be better to change “high and low levels of adiponectin…” to “altered levels of adiponectin”
- Introduction:
- Perhaps add a modifier to the second clause of the sentence in Line 32 in order to emphasize how unfavorable dietary habits, lifestyle etc. Are still the most important factors but genetics also play a role. I believe this is helpful since the conjunction “but” is used: ...psychosocial factors are most important, but endocrine disorders and genetic disposition also have an identifiable contribution to obesity.
- Line 34. Authors may want to describe “genetic predisposition” NOT “genetic disposition”
- Line 35 change “Ghrelin and the adipokines leptin and adiponectin…” to “Ghrelin, the adipokines leptin and adiponectin”.
- Line 48. The authors stated incorrect findings of [10] “A recently published study reported associations with 37 outcomes representing 18 distinct disorders [10]. APOE É›4É›4 was associated with reduced prevalence of obesity (OR = 0.78), and É›2É›2 with increased prevalence of obesity (OR = 50 1.22) compared with É›3É›3”. It should be “APOE É›4É›4 was associated with increased prevalence of obesity (OR = 0.78), and É›2É›2 with decreased prevalence of obesity (OR = 50 1.22) compared with É›3É›3”, since overall É›4 is risk allele and É›2 is protective allele for obesity and many other diseases…
- Line 55 change “We showed in a previous study..” to “In a previous study we demonstrated an association….”
- Material and method:
- Line 71, comma can be removed after “psychiatric disorders”
- Line 73: In the first six months after inclusion, the...
- Line 113, it should be” The APOE alleles or APOE polymorphisms, were determined” not “The APOE alleles polymorphism was determined” since alleles are different forms of gene locus, such as APOE locus.
- Line 113, if it is not typo, what rational of these three groups? “three groups: E2 (É›2É›2 + É›2É›3), E3 (É›3É›3 + É›2É›4), and E4 (É›3É›4 + É›4É›4)”. Why you grouped É›2É›4 into E3 group since one expects E3 group should include people who carry e3 allele (e.g., É›3É›4, É›3É›2 or É›3É›3). It might be better either exclude this subject with É›2É›4 or into either E2 or E4 group
- Author should provide reference here after “in a recently submitted paper by Jansen A. et. Al”.
- Line 118. Author should change “e2ε4” to “ε2ε4”, please keep consistent through entire manuscript once descripted APOE alleles
- Line 120: extra space after parenthesis before ɛ3ɛ3 + ɛ2ɛ4), and E4..., should be (ɛ3ɛ3 + ɛ2ɛ4), and E4
- Line 134. Is it typo here in underline of ad? “or multiple testing ad modum Benjamini-Hochberg…”
- Result
- Line 140 should have something like “Sixty-two participants who had complete data before and 12 months after surgery, were included in this study”.
- Discussion:
- Line 166 change to “In our current study, the main findings include the positive and negative”
- Line 170, repetitiveness with “appetite-regulating effect”. Sentences can be simplified to: “Adiponectin is a hormone that passes into the brain and has an appetite-regulating effect, increases energy expenditure...”
- Line 176. Authors have incorrect statement here “The positive association between E4 and adiponectin in this study is in accordance with the protective effect of É›4 against obesity [10]” since based on your study (tables and figure) and [10] which show “The positive association between E4 (É›3É›4, É›4É›4) and adiponectin in this study is in accordance with É›4 allele is risk for obesity…”
- Similar issue in line 181, “The significantly higher HbA1c values in the group E4 compared with group E3 is not as expected if É›4 protects against diabetes, and is not as reported by Lumsden et al. [10]” since [10] shows “as expected, ε3ε4 and ε4ε4 genotypes associated with increased odds of diabetes” after we checked this publication. Thus, authors should change “… É›4 is risk allele for diabetes”
- Line 187. Authors may want to provide possible explanations for discrepancies among different studies such as, sample size, different ethnics or study design, “present study population, (i.e. subjects with morbid obesity), might contribute to discrepancies between this and other…”
- Line 199. Most studies show E4 group (ε3ε4 and ε4ε4) positive associations with obesity (or dementia), and cortisol level. These studies include “Review Domest Anim Endocrinol. 2016 Jul;56 Suppl:S112-20. doi: 10.1016/j.domaniend.2016.03.004. Epub 2016 Mar 31.Stress, cortisol, and obesity: a role for cortisol responsiveness in identifying individuals prone to obesity, however your study show an inversed (or negative) association between E4 and cortisol level, “The negative association between the E4 group and cortisol seen in this study indicates that the protective effect of É›4 against obesity…”. Thus authors should discussed discrepancies between this study and others.
- Line 208. It might be better to make changes to “In accordance with other studies, no statistically significant associations of APOE-polymorphism with thyroid, parathyroid disorders, sex hormones, the adipokines ghrelin and leptin were observed [10]”
- Line 212. Authors should provide a full name for CRP, then abbreviation
Reviewer 2 Report
The present study explored associations between APOE polymorphism and endocrine functions in subjects with obesity undergoing bariatric surgery. Apolipoprotein E alleles were grouped and analyzed with a linear mixed model to predict the hormonal effects of the APOE groups. This is a retrospective case-series study with a group of subjects referred for evaluation of bariatric surgery. The treatment followed international guidelines with a lifestyle intervention before surgery and hormones known to be related to obesity were analyzed.
High clinical importance may count as a worthy preliminary study to be confirmed with a bigger and better-designed prospective study.
Major issues:
small ample size. The sample is small weakening the already not optimal retrospective case series design. The grouping strategy to increase size per group may mask some of the effects. grouping of APOE2–4 heterozygote individual seems arbitrary; no clear explanation of its assignment to E3 group is provided. A different grouping strategy or exclusion of the e2e4 individual for analysis may help reduce type II error.
Adiponectin concentrations in plasma are higher in females compared to males. As stated in the limitation section, unbalanced gender distribution in favor of females limits the external validity. However, even after adjustment for sex effects, results may represent a female-specific effect of APOE impossible to confirm with current data.
Case-control series would have been preferable for better correlations in obesity: healthy adults without intervention or dietary intervention. Since the study is retrospective, a reference or more information regarding APOE polymorphisms and association with adiponectin and cortisol in healthy individuals is necessary.
Minor :
No associations with body mass index indicate effects on metabolic functions but not direct effects on bodyweight as stated in conclusions. Although metabolic function regulates BMI, based on results presented, change in weight is more likely an APOE-independent effect and as it is, the statement, in conclusion, sounds causative.
Do the authors have information on the lipid profiles of patients?. Association with adiponectin and cortisol with APOE levels would confirm associations found with APOE genotype.
Discussion over lack of association of leptin and ghrelin with APOE genotype, other than confirmation with other studies is lacking.
Round 2
Reviewer 2 Report
The authors answered all comments and concerns and revised the manuscript accordingly. No major changes in data analysis or additional data were presented. However, they covered it on limitations or gave good reasoning on their choices. A few minor revisions are needed.
Please add the provided explanation in line 131 "This is in accordance with the observation that for many clinical or biochemical properties investigated and published(cite references), the ɛ2 and ɛ4 alleles have opposite effects relative to the ɛ3 allele. Therefore it makes sense in a clinical setting to group ɛ2ɛ4 alleles phenotypically as E3 although the ɛ2ɛ4 genotype is not genetically ɛ3."
regarding Answer " The available information about the lipid profiles was unfortunately not judged as suitable for use in this paper. " Do the authors have or do not have access to these data? Who considered not suitable? Perhaps adding a statement on the limitations section mentioning that the available data could confirm the associations found with cortisol and adiponectin would be desirable.
Please change lines 289 and 290, in conclusion, to better reflect the direct APOE-dependent effects "indicating effects on body weight and metabolic functions." to something like... indicating effects on metabolic functions and thus body weight.
Author Response
Please see the attachment.

This manuscript is a resubmission of an earlier submission. The following is a list of the peer review reports and author responses from that submission.
Round 1
Reviewer 1 Report
APOE polymorphism and endocrine functions in subjects with morbid obesity undergoing bariatric surgery. This manuscript is well designed and comprehensive, but I have some minor issues: 1. What is the pathogenesis of APOE variants with Morbid obesity?2. In line 107, the authors subdivided genotyping of APO E variants into E2, E3, and E4, but E3 (ε2ε2 + ε2ε4), how come???? I suggested to subdivide into E2 (ε2ε2 + ε2ε3 + ε2ε4), E3 (ε3/ε3), E4 (ε3/ε4 + ε4/ε4).
3. The results should be reconstructed according to the new parameters with E3 carriers as reference.
4. Some bioinformatics tools such as string database, cellular compartments, and GeneMania could be helpful in explaining the molecular function of this gene.
Reviewer 2 Report
1. “APOE” should be corrected throughout the text. When the authors indicate APOE gene it should be written in italic.
2. In abstract and methods (Line 20 and 108), it is stated that E3 group includes ɛ3ɛ3+ɛ2ɛ4. Is it a good classification to include ɛ2ɛ4 in this group? The combination of ɛ2 and ɛ4 isoforms usually excluded because of the adverse impacts.
3. Total number of individuals recruited in the study should be stated in “study design and participant” section.
4. There are some concerns about the sample size of the study. The number of individuals in the ApoE groups are very low.
Round 2
Reviewer 1 Report
All modifications have been done